# Quantifying the sources of increasing stratospheric water vapour concentrations

Patrick E. Sheese[1], Kaley A. Walker[1], Chris D. Boone[2], and David A. Plummer[3]

[1]University of Toronto, Department of Physics, Toronto, Canada
[2]University of Waterloo, Department of Chemistry, Waterloo, Canada
[3]Candian Centre for Climate Modelling and Analysis, Environment and Climate Change Canada, Montreal, Canada

**Correspondence:** Kaley A. Walker (kaley.walker@utoronto.ca)

**Abstract.** According to satellite measurements from multiple instruments, water vapour ($H_2O$) concentrations, in most regions of the stratosphere, have been increasing at a statistically significant rate of $\sim$1-5% dec$^{-1}$ since the early 2000s. Previous studies have estimated stratospheric $H_2O$ trends, but none have simultaneously quantified the contributions from all main sources (temperature variations in the tropical tropopause region, changes in the Brewer-Dobson circulation, and changes in methane ($CH_4$) concentrations and oxidation) at all latitudes. Atmospheric Chemistry Experiment – Fourier Transform Spectrometer (ACE-FTS) measurements are used to estimate altitude/latitude-dependent stratospheric $H_2O$ trends from 2004-2021 due to these sources. Results indicate that rising temperatures in the tropical tropopause region play a significant role in the increases, accounting for $\sim$1-4% dec$^{-1}$ in the tropical lower-mid stratosphere, as well as in the mid-latitudes below $\sim$20 km. By regressing to ACE-FTS $N_2O$ concentrations, it is found that in the lower-middle stratosphere, general circulation changes have led to both significant $H_2O$ increases and decreases on the order of 1-2% dec$^{-1}$ depending on altitude/latitude region. Making use of measured and modelled $CH_4$ concentrations, the increase in $H_2O$ due to $CH_4$ oxidation is calculated to be $\sim$1-2% dec$^{-1}$ above $\sim$30 km in the Northern Hemisphere and throughout the stratosphere in the Southern Hemisphere. After accounting for these sources, there are still regions of the midlatitude lower-mid stratosphere that exhibit significant residual $H_2O$ trends increasing at 1-2% dec$^{-1}$. Results indicate that these unaccounted for increases could potentially be explained by increases in upper tropospheric molecular hydrogen.

## 1 Introduction

Water vapour ($H_2O$) is the most abundant greenhouse gas in the Earth's atmosphere. Much like other greenhouse gases, it absorbs shortwave radiation near the surface, leading to temperature increases, and emits longwave radiation in the stratosphere and above, leading to upper atmospheric cooling. Near the surface, $H_2O$ is part of a positive feedback loop where increasing temperatures lead to an increase in the saturation vapour pressure, leading to more $H_2O$ in the atmosphere, leading to more heating, and so on. Because $H_2O$ is controlled predominantly by this feedback system in the troposphere, and due to its much shorter atmospheric lifetime (with respect to other greenhouse gases) on the order of weeks near the surface and 10-20 years in the stratosphere (Brasseur and Solomon, 2005), $H_2O$ is typically considered an amplifier of the greenhouse effect rather than a contributor (e.g., Chung et al., 2014). Downward propagating radiation from stratospheric $H_2O$ can also lead to upper

tropospheric heating (e.g., Manabe and Wetherald, 1967; de F. Forster and Shine, 1999; Forster and Shine, 2002). It is therefore important to continually monitor and understand $H_2O$ variations throughout the troposphere and stratosphere.

Although the predominant source of stratospheric $H_2O$ is moisture-rich tropospheric air that is lofted upwards in the tropics as part of the Brewer-Dobson circulation (BDC), the physical processes that control the amount of water vapour in the stratosphere are fundamentally different from those in the troposphere. As that moisture-rich tropospheric air crosses the cold

tropical tropopause region, it freeze-dries, removing most of the $H_2O$ before entering the stratosphere (Brewer, 1949; Holton and Gettelman, 2001, and references therein). With few other sources, and its relatively long stratospheric lifetime, much of the stratospheric $H_2O$ budget is a function of tropical tropopause cold-point temperature, especially in the lower stratosphere. As such, time series of tropical tropopause region temperatures are often used as a regressor in determining stratospheric $H_2O$ trends (Hegglin et al., 2014, and references therein). Hegglin et al. (2014) showed that variations in low-mid latitude, lower

stratospheric $H_2O$ very closely followed variations in modelled mean tropical temperatures at 100 hPa. At higher altitudes and more poleward latitudes, $H_2O$ variations tended to follow those of the modelled temperatures with a lag of a few months (tape recorder effect (Mote et al., 1996)).

Another major source of stratospheric $H_2O$ is $CH_4$ oxidation via reactions with OH, $O(^1D)$, and Cl. As detailed in Brasseur and Solomon (2005), oxidation of $CH_4$ via OH can produce $H_2O$ directly, but all three reactions have byproducts that lead to

the production of a formaldehyde molecule ($CH_2O$), which is then quickly destroyed via multiple reactions that can produce $H_2O$ molecules. In the stratosphere, on average, an oxidized $CH_4$ molecule produces $\sim 2$ $H_2O$ molecules, however that average varies with altitude and latitude (e.g., Jones et al., 1986; le Texier et al., 1988; Frank et al., 2018).

A minor source of stratospheric $H_2O$ is the oxidation of $H_2$, which can be an indirect product of $CH_4$ oxidation or transported into the stratosphere from the tropical troposphere. Both Wrotny et al. (2010) and Frank et al. (2018) have shown that it is

possible for the ratio of $H_2O$ production to $CH_4$ loss, $\alpha$, in the tropical stratosphere to be greater than 2. Wrotny et al. (2010) used satellite measurements from HALOE (Halogen Occultation Experiment), ACE-FTS (Atmospheric Chemistry Experiment – Fourier Transform Spectrometer), and MIPAS (Michelson Interferometer for Passive Atmospheric Sounding) to show that $\alpha$ can be on the order of 2.0-3.7, attributing the additional production to oxidation of $H_2$ that was not produced via $CH_4$ oxidation.

A number of studies have recently been conducted that have measured stratospheric $H_2O$ increases from satellite measure-

ments, however, none of them parse trends in order to determine the relative contributions from each of the three main sources throughout the stratosphere. For instance, similar to Hegglin et al. (2014) and Tao et al. (2023), Randel and Park (2019) used merged HALOE and Aura/MLS (Microwave Limb Sounder) data to show that the majority of variations in lower stratospheric $H_2O$ from 1993-2017 can be explained by changes in tropical cold point temperature. Yue et al. (2019) determined that both SABER (Sounding of the Atmosphere using Broadband Radiometry) and Aura/MLS measurements exhibited stratospheric

$H_2O$ trends on the order of 5-6% $dec^{-1}$ since the early 2000's. As to the sources of those increases, they determined that the $H_2O$ trends in the lower stratosphere are consistent with frost-point hygrometer measurements, and mesospheric trends are much greater than what is expected assuming complete $CH_4$ oxidation. Similarly, Fernando et al. (2020) found that profiles of global ACE-FTS, Aura/MLS, and SABER $H_2O$ trends agreed well, but could not be explained solely by $CH_4$ oxidation. ACE-FTS is in a unique position when it comes to investigating the influence of $CH_4$ oxidation on stratospheric $H_2O$ trends as

it is currently the only Earth observing satellite instrument that makes vertically resolved measurements of both $H_2O$ and $CH_4$ throughout the stratosphere.

This study uses simultaneously-measured profiles of $H_2O$, $CH_4$, and $N_2O$ from ACE-FTS (a measurement combination that only ACE-FTS is currently producing) in order to measure height resolved $H_2O$ trends throughout the stratosphere in latitudinal bands spanning 80°S-80°N. The sources of those trends are then quantified, considering contributions due to temperature changes in the tropical tropopause region, structural changes in the BDC, and changes in local and tropical tropopause region $CH_4$ concentrations.

A description of the satellite measurements used in this study can be found in Section 2, and the methodology is described in Section 3. The ACE-FTS $H_2O$ trends and the contributions from different sources are discussed in Section 4, and all the results are summarized in Section 5.

## 2 ACE-FTS on SciSat

The ACE-FTS instrument (Bernath et al., 2005) is one of two instruments on board the Canadian SciSat satellite, which was launched into a high inclination orbit in August 2003. Starting in February 2004, ACE-FTS began measuring profiles of temperature, pressure, and concentrations of multiple atmospheric trace species, including $H_2O$, $CH_4$, and $N_2O$. The instrument is a high-spectral-resolution ($0.02$ cm$^{-1}$) spectrometer viewing the Earth's limb in the infrared between 750 and 4400 cm$^{-1}$, using solar occultation viewing geometry. The vertical profiles span 5-150 km with a vertical spacing of ∼2 to 6 km, depending on the orbital geometry, and the circular field-of-view at the tangent altitude is on the order of 3-4 km. This study makes use of the most recent version of level 2 data, version 5.2 (v5.2), which provides interpolated data on a 1-km grid. The retrieval algorithm, described by Boone et al. (2005, 2013, 2020, 2023) uses a non-linear, least-squares, global-fitting technique that fits observed atmospheric transmission spectra to forward modelled spectra in species/altitude dependent microwindows. The modelled spectra are calculated using spectral line parameters from the HITRAN2020 database (Gordon et al., 2022). In all retrievals, horizontal homogeneity is assumed and diurnal variations are not taken into account along the line-of-sight.

Version 5.2 of the $H_2O$ retrievals makes use of 63 microwindows between 937 and 3173 cm$^{-1}$, has altitude limits of 5 and 95 km, and in the stratosphere accounts for $CO_2$, $O_3$, $N_2O$, $CH_4$, $NO_2$, $HNO_3$, $NO$, and $COF_2$, as well as isotopologues $H_2O$, $CO_2$, $N_2O$, and $CH_4$ as interfering species. ACE-FTS $H_2O$ has been used in over 50 different studies, including the European Space Agency's Water Vapour Climate Change Initiative (Hegglin and Ye, 2022; Ye et al., 2022), merged data studies (e.g., Froidevaux et al., 2015; Davis et al., 2016), and multiple validation studies (e.g., Wetzel et al., 2013; Weaver et al., 2019; Rong et al., 2019). Fernando et al. (2020) examined ACE-FTS v4.0 $H_2O$ and $CH_4$ trends in the stratosphere and mesosphere, however focused on 55°S-55°N mean values. Although the study did not quantify the different sources contributing to $H_2O$ trends, it was concluded that increasing $CH_4$ trends were not sufficient to fully explain the observed increases in stratospheric $H_2O$.

The latitudinal coverage of the instrument is shown in Fig.1. As can be seen, the annually-repeating measurements are made predominantly in the high latitudes, with the tropics being sampled roughly every three months. However, significant trends are still capable of being detected at the lower latitudes due to the long lifetime of the ACE-FTS data sets.

## 3 Methodology

Many different trends are calculated in this study, each of them making use of the multiple linear regression (MLR) technique (Chatterjee and Hadi, 1986), using various predictor data sets (in different combinations) as regressors. In each case; whether it is ACE-FTS $H_2O$, $CH_4$, or $N_2O$ trends; the time series are fit to a model of the form,

$$y_{fit} = \beta_0 + \beta_1 l(t) + \sum_i \beta_i r_i(t), \tag{1}$$

where t is time in years, $\beta$ are the fit components, $l(t)$ is a linear function increasing from -0.5 to 0.5 over the length of the time 100 series being fitted, and $r(t)$ are the considered regressor time series. The regressor time series used in this study are,

- two annual oscillation terms $\cos 2\pi t$ and $\sin 2\pi t$ (AO) and two semi-annual oscillation terms $\cos 4\pi t$ and $\sin 4\pi t$(SAO), with $t$ measured in years;

- monthly mean tropical tropopause region temperatures from ECMWF (European Centre for Medium-Range Weather Forecasts) Reanalysis version 5 (ERA5 (Hersbach et al., 2020)) data (as described below);

- simultaneously measured ACE-FTS $N_2O$ data as a proxy for dynamical processes (Dubé et al., 2023). As per Dubé et al. (2023), local $N_2O$ time series at all altitudes and latitudes have a 2.8% $dec^{-1}$ trend (corresponding to the 2004-2022 global surface $N_2O$ trend) removed prior to being used as a regressor;

- daily mean F10.7 cm solar radio flux values (which indirectly affect $H_2O$ concentrations via influences on $O_3$ and temperature) provided by Geomagnetic Observatory Niemegk, Potsdam (Matzka et al., 2021);

- monthly mean Quasi-Biennial Oscillation (QBO) proxies of the 30 and 50 hPa Singapore zonal winds (Baldwin et al., 2001, obtained from https://www.geo.fu-berlin.de/en/met/ag/strat/produkte/qbo/index.html, last access: 30 Sept 2023));

- monthly mean El Niño/Southern Oscillation (ENSO) index values from the NOAA Physical Sciences Laboratory (Wolter and Timlin, 2011, obtained from https://psl.noaa.gov/enso/mei/, last access 30 Sept 2023);

- monthly mean tropopause pressure (trop) values from NCEP-NCAR reanalysis data (Kalnay et al., 1996, obtained from 115 ftp://cdc.noaa.gov/, last access 10 Jan 2025).

$H_2O$ trends were calculated at all ACE-FTS altitudes (1-km grid) between 17.5 and 55.5 km—roughly between the hygropause and stratopause—in sixteen $10°$ bins between $80°S$ and $80°N$, for daily-mean time series. To avoid influences from measurements within and near the polar vortexes, scaled potential vorticity (sPV) values derived from the Modern Era Retrospective

analysis for Research and Applications, Version 2 (MERRA-2 (Gelaro et al., 2017)) interpolated to ACE-FTS locations (Manney et al., 2007) were employed. Only data with a corresponding absolute sPV value of $1.4 \times 10^{-4}$ s$^{-1}$ or less were used in this study.

In order to fit to tropical tropopause region temperatures, a time series of monthly mean temperatures within 15°S-15°N at 100 hPa for years 1988-2022 was obtained from ERA5 data (shown in Fig. 2). In the tropical lower stratosphere, it is expected that the $H_2O$ time series would closely follow the ERA5 temperature time series. However, at locations further from the tropical lower stratosphere, the $H_2O$ response is expected to be lagged with respect to the temperature time series, as it takes longer for air entering the stratosphere to reach those locations, as discussed above. In the fitting algorithm, at each altitude and latitude bin, the regression was performed with the ERA5 temperatures lagged by 0-15 years in 2-day increments to find the lag time that minimized the residual between the ACE-FTS $H_2O$ data and the MLR fit. Only lag times that led to a positive correlation between the $H_2O$ and temperature time series were considered, and the lagged temperature time series will be referred to hereafter as $T_{lag}$. At the lowest altitude level in the 0-10°S and 0-10°N latitude bins, lag times were restricted to within 2 months, and lag times at any other given altitude/latitude bin were restricted to a value within $\pm 24$ months of adjacent bins. Although the ERA5 time series was lagged by up to 15 years in each altitude/latitude bin, it was found that the maximum lag time required to minimize the residuals in any bin was 60 months, whereas stratospheric mean age-of-air estimates tend to be on the order of 0-15 years, depending on altitude and latitude. This can be explained by the fact that lagging the temperature time series assumes that the measured air parcel is a singular parcel that traveled from the entry point to the measurement location with a particular transit time and does not account for mixing or changes in transit pathways (Poshyvailo-Strube et al., 2022).

To ensure that it is appropriate to simultaneously use $T_{lag}$ and local $N_2O$ time series as regressors, the correlation between these time series was calculated in each altitude and latitude bin. At all altitudes and latitudes, the absolute correlation between measured $N_2O$ and $T_{lag}$ time series is less than 0.35 and is typically below 0.2. The same is true for the correlation between $N_2O$ and the seasonal cycles, and for $T_{lag}$ and SAO–although between $T_{lag}$ and SAO the correlation is typically on the order of 0.2-0.3. $T_{lag}$ and AO, however, are not independent as the tropopause region temperatures exhibit a significant annual oscillation. The implications of this are discussed in Sect. 4.2.

Although $CH_4$ oxidation is a major source of stratospheric $H_2O$, ACE-FTS measurements of local $CH_4$ concentrations are not an appropriate regressor, as local $H_2O$ concentrations depend on the amount of $CH_4$ that has been oxidized in the air parcel since entering the stratosphere, which is a function of the difference between the $CH_4$ concentration at time of entry and the local $CH_4$ concentration,

$$[H_2O]_{CH_4} = \alpha[CH_4]_{oxidized} = \alpha([CH_4]_{entry} - [CH_4]_{local}), \tag{2}$$

where $\alpha$ is the $H_2O$ yield from oxidized $CH_4$. In past studies, $\alpha$ is often assumed to be a constant of 2 throughout the stratosphere (e.g., Stowasser et al., 1999; Myhre et al., 2007; Frank et al., 2018). However, Frank et al. (2018) showed that this assumption tends to overestimate $H_2O$ production in the lower stratosphere and underestimate $H_2O$ production nearer the stratopause. In this study, a height dependent $\alpha$ is used based on the global effective $H_2O$ yield profile shown in Fig. 14 of

Frank et al. (2018), which is ∼1.6 in the lower stratosphere and ∼2.2 at the stratopause. To account for the fraction of $H_2O$ trends due to $CH_4$ oxidation, the time derivative of Eq. 2 is taken,

$$\frac{d[H_2O]_{CH_4}}{dt} = \alpha\Big(\frac{d[CH_4]_{entry}}{dt} - \frac{d[CH_4]_{local}}{dt}\Big).$$ (3)

The local $CH_4$ trends are determined by regressing to ACE-FTS $N_2O$ data (in addition to AO and SAO time series) to account for changes in $CH_4$ due to changes in the general circulation,

$$\frac{d[CH_4]_{local}}{dt} = \beta_0 + \beta_1 l(t) + \beta_{AO}^{(2)} AO + \beta_{SAO}^{(2)} SAO + \beta_{N_2O}[N_2O].$$ (4)

Since ACE-FTS has low sampling in the tropical region, model data from the specified dynamics run of the Canadian Middle Atmosphere Model (CMAM39-SD) (Beagley et al., 1997; Scinocca et al., 2008; McLandress et al., 2014) were used to supplement the ACE-FTS data when calculating $CH_4$ entry trends. The CMAM39-SD run (referred to as CMAM39 hereafter) spans 1979-2018 inclusive, with simulations relaxed towards six-hourly fields of temperature, vorticity, and divergence from ERA-Interim (Dee et al., 2011) reanalysis data. The chemical forcing fields for long-lived greenhouse gases, including $CH_4$, were obtained from the Coupled Model Intercomparison Project Phase 6 (CMIP6) (Eyring et al., 2016)) historical time series (Meinshausen et al., 2017) up to 2014, and the SSP2-4.5 scenario (Meinshausen et al., 2020) for the remaining years. They were forced as a time-dependent mixing ratio specified for the bottom two model layers (approximately 100 m in depth) based on the global and annual average mixing ratio taken as the mid-year value and linearly interpolated in time to provide values at intermediate times.

The mean of the ACE-FTS 15°S-15°N, 100-200 hPa, 180-day running zonal mean and the CMAM39 15°S-15°N, 100-200 hPa, daily zonal mean was calculated, shown in Fig. 2. This ACE-CMAM mean time series (black dashed line in Fig. 2) was used to determine a $CH_4$ entry trend of 78±1 ppbv/dec between 2004 and 2022. The uncertainties for $\frac{d[CH_4]_{entry}}{dt}$ and $\frac{d[CH_4]_{local}}{dt}$ are added together in quadrature, as are the uncertainties in $\frac{d[H_2O]_{CH_4}}{dt}$ and the fitted $H_2O$ trend uncertainties (all uncertainties are the statistical uncertainties of the calculated trends, excluding measurement uncertainties, which are assumed to be negligible). This method however assumes that $CH_4$ trends at the stratospheric entry point have been constant from the time of entry to the time period for which the local $CH_4$ trends are being calculated, which could be a difference of up to the order of a decade. As seen in the CMAM $CH_4$ time series, and as discussed by, e.g., Dlugokencky et al. (2003) and Rigby et al. (2008), there was a slowdown in the increase of $CH_4$ concentrations just before the beginning of the ACE mission (∼1999-2003) just below the tropical tropopause. This affect is accounted for and discussed when examining $H_2O$ trends using time-lagged $CH_4$ entry trends in Sect. 4.2.

All ACE-FTS data were screened for outliers using data quality flags, as per Sheese et al. (2015), prior to analysis. At all altitude levels used in this study, the screening rejects less than 1% of the data. Only data prior to 2022 were used as to avoid any influence from $H_2O$ injected by the Honga Tonga-Honga Ha'apai eruption in January 2022.

## 4 Results

The following sections discuss results of MLR trend analysis on the ACE-FTS $H_2O$ time series using different regressors. The "full" $H_2O$ trends are those where only the AO and SAO time series (that do not themselves have any trend) are used as regressors. Results are also shown for $H_2O$ "residual" trends, which are the resulting trends when using additional time series that may contain a trend as regressors. The differences between the full trends and the residual trends (labelled as $\Delta$ in figure panels) are considered to be the contribution to the full trend due to the regressors used in the analysis.

Section 4.1 discusses the full $H_2O$ trends as well as the individual contributions to the full $H_2O$ trends due to solar flux, QBO, and ENSO influences; structural BDC changes; and changes in tropical tropopause region temperatures. Section 4.2 analyzes the contribution to the full $H_2O$ trends due to $CH_4$ oxidation under two different assumptions: first, simply assuming a constant $\frac{d[CH_4]_{entry}}{dt}$ over the past few decades and then accounting for the fact that $\frac{d[CH_4]_{entry}}{dt}$ has not been constant and allowing the value used to vary depending on altitude and latitude. The discussion then focuses on the remaining residual trends, which can still be statistically significant, after all the above-mentioned individual sources contributing to the full trends are accounted for.

### 4.1 Standard MLR Results

A simple MLR analysis of the ACE-FTS $H_2O$ trends, regressing only to annual and semi-annual cycles, are shown in the left panel of 3 and the corresponding trend uncertainties (95% confidence level) are shown in the right panel. The results clearly show that since 2004 stratospheric $H_2O$ has been increasing, or has had no significant trend, throughout the stratosphere. There is a noted hemispheric asymmetry at all altitudes, except for around the highest altitudes, $\sim$50-55 km, where trends are on the order of 2-3%. In the Southern Hemisphere (SH), $H_2O$ trends also tend to be on the order of 2-3%, except near the tropical lower stratosphere. In the Northern Hemisphere (NH), $H_2O$ trends tend to be somewhat more variable, with values of 3-5% $dec^{-1}$ up to $\sim$30 km and $\sim$1-3% $dec^{-1}$ above 30 km.

The left panel of Fig. 4 again shows the trend results from a simple MLR analysis of the ACE-FTS $H_2O$ time series, regressing only to annual and semi-annual cycles, and these are shown to compare to the residual $H_2O$ trends when regressing to the F10.7 cm, QBO, ENSO, and trop time series, shown in the middle panel. The differences in the $H_2O$ trend values are typically less than $\pm$0.5% $dec^{-1}$ in all altitude/latitude bins (except within 70-80°S), and the differences due to F10.7 cm alone are typically only within $\pm$0.2% $dec^{-1}$ (not shown). The effects of time-lagging ENSO and tropopause pressure are discussed in Appendix A1.

In all following analyses, QBO, ENSO, and trop indicies are not used in the regression schemes as the effects they are meant to represent can also be accounted for by the ACE-FTS $N_2O$ time series.

As shown in Dubé et al. (2023), simultaneous measurements of $N_2O$, a long-lived atmospheric tracer, can be used as a proxy for changes in the BDC and can be used as an alternate regressor to account for trends due to dynamical processes in the stratosphere. The middle panel of Fig. 5 shows the residual $H_2O$ trends after regressing to local ACE-FTS $N_2O$, and the right panel shows the difference between the full $H_2O$ trend and those after regressing with $N_2O$. The remaining trends are all

increasing (Fig. 5, middle panel) but the hemispheric asymmetry in the lower stratosphere flips sign and the remaining trends are more strongly positive in the SH ($\sim$3% dec$^{-1}$) compared to the NH (1-3% dec$^{-1}$). Above $\sim$35 km the remaining H$_2$O trends are more consistent between hemispheres, on the order of 1-3% dec$^{-1}$. These results are consistent with those of Tao et al. (2023). The differences (right panel of Fig. 5) represent the trend in H$_2$O due to changes in the general circulation, showing that the net hemispheric asymmetry in H$_2$O trends can be attributed to changes in stratospheric circulation. In particular, the influence can be to either increase or decrease local H$_2$O concentrations, depending on the region. Changes in the BDC account for an increase in H$_2$O of 1-2% dec$^{-1}$ in the NH near 20-30 km and a decrease in H$_2$O of 1-2% dec$^{-1}$ in the NH near 30-40 km as well as in the SH near 25-30 km. In all other regions, the contribution of dynamical processes to H$_2$O trends is not statistically significant.

Climate models have long indicated that increasing concentrations of greenhouse gases in the lower atmosphere should lead to an acceleration of both the shallow and deep branches of the BDC (e.g., Butchart, 2014, and references therein). Multiple studies have examined measurements of atmospheric proxies for BDC changes and have detected accelerations in the shallow branch (below 20 km), although not at higher altitudes in the deep branch (e.g., Engel et al., 2009; Diallo et al., 2012; Engel et al., 2017). However, recent studies have suggested that decreasing concentrations of ozone depleting substances in the stratosphere can lead to a deceleration of the BDC (e.g., Polvani et al., 2018; Fu et al., 2019). Polvani et al. (2018) analyzed data from WACCM runs for 1965-2080 and showed that between 2000 and 2080, the BDC is expected to slow down due to the removal of ozone depleting substances. Fu et al. (2019) show that within 2000-2018 there has been a slowing down of the BDC in the SH lower stratosphere (10-50 hPa), but no clear overall trend in the NH. This agrees with ACE-FTS showing a decrease in H$_2$O due to structural circulation changes throughout most of the SH and a combination of increasing and decreasing circulation in the NH depending on the season. Since Fu et al. (2019) averaged data over the entire NH from 10-50 hPa, it is unknown whether the spatial patterns of the seasonal differences they reported would be in agreement with the BDC results in this study.

Since it can take months to years for newly introduced stratospheric air (in the tropical lower stratosphere) to be transported throughout the stratosphere, including time-lagged ERA5 tropical upper-tropospheric temperatures ($T_{lag}$) in the regression has a significant effect on the trend results. Including $T_{lag}$ in the regression can decrease the residual trends by up to 4% dec$^{-1}$, as seen in Fig. 6, indicating a warming trend near the tropical tropopause, which would allow more H$_2$O to enter the stratosphere. In the tropics, this warming is contributing a $\sim$2-4% dec$^{-1}$ increase in H$_2$O below 20 km, and a $\sim$1-2% increase in the mid stratosphere up to about 45 km (corresponding to a reduction of the residual H$_2$O trends). The warming also contributes a 1-3% increase in H$_2$O in the mid-latitude lower stratosphere (below $\sim$20 km). Elsewhere, including $T_{lag}$ as a regressor does not significantly affect H$_2$O trends, with differences from the full trend typically within $\pm$1%. Figure 7 shows the lag times that were determined to minimize the difference between the fit and the ACE-FTS data. As expected, the lag times increase with altitude and with absolute latitude, as stratospheric age of air increases. The lag times are on the order of a 1-2 months near the equator in the lower stratosphere and increase up to 3-5 years nearer the high-latitude stratopause regions.

## 4.2 Accounting for CH$_4$ oxidation

In order to quantify how much CH$_4$ oxidation is contributing to stratospheric H$_2$O trends, first local ACE-FTS CH$_4$ trends were calculated using an annual cycle, a semi-annual cycle, and ACE-FTS N$_2$O time series as regressors. As seen in the left panel of Fig. 8, the CH$_4$ trends are increasing in all regions and also exhibit a significant hemispheric asymmetry. In the mid-high latitudes, NH CH$_4$ trends range from 3% dec$^{-1}$ in the lower stratosphere up to 12% dec$^{-1}$ near 55 km. These trends are greater than the SH trends that increase from 2% dec$^{-1}$ up to 8% dec$^{-1}$. At the lower latitudes, hemispherical differences are only on the order of 1-2% dec$^{-1}$, with relatively larger trends in the NH around 20-30 km and relatively larger trends in the SH around 40-55 km. The right panel of Fig. 8 shows how those trends contribute to the stratospheric H$_2$O via Eq. 3. The increases in CH$_4$ concentrations are leading to an increase of the H$_2$O budget of $\sim$1-3% dec$^{-1}$ above $\sim$35 km, $\sim$1-2% dec$^{-1}$ below 35 km, with insignificant influence closer to the tropical tropopause region where there is expected to be little to no contribution from methane oxidation.

As shown in Fig. 9, when $[H_2O]_{CH_4}$ trends are subtracted from the residual H$_2$O trends that are calculated regressing to AO, SAO, N$_2$O, $T_{lag}$, and F10.7 cm time series, most of the trends throughout the stratosphere are within $\sim \pm 1\%$ dec$^{-1}$ and are not statistically significant. This indicates that these regressors can account for the full ACE-FTS H$_2$O trends throughout the majority of the stratosphere. The exceptions are in the mid to high latitude regions ($\sim$30-70°S and 40-70°N) in the lower-mid stratosphere ($\sim$20-35 km). In these regions there are still significant residual H$_2$O trends of $\sim$1-2% dec$^{-1}$. However, $\frac{d[CH_4]_{entry}}{dt}$ has not been constant over the past 20-30 years, as shown in Fig. 2. To account for this, a time-dependent trend analysis was performed on the CH$_4$ entry time series. Lagged 18-year trend values for the CMAM-ACE CH$_4$ entry time series were calculated for lag times of 0-10 years in 5-day intervals (i.e., a lag value of 10 years corresponds to the trend for 1994-2012). The calculated trends versus lag times are shown in Fig. 10, and, as can be observed in Fig. 2, the 18-year CH$_4$ entry trends have been increasing since the early 1990's. This method of using lagged $\frac{d[CH_4]_{entry}}{dt}$ was employed by Hegglin et al. (2014). That study used time-lagged $\frac{d[CH_4]_{entry}}{dt}$ with lag times corresponding to the local mean age-of-air. However, ACE-FTS does not have validated measurements of age-of-air throughout the stratosphere, and therefore the optimal lag times from the temperature regression (Fig. 7) are used as a proxy for mean age-of-air.

As expected, the H$_2$O trends due to CH$_4$ trends that account for time lags (Fig. 11) are less than those that use a constant CH$_4$ entry trend value (Fig. 8, right panel). Throughout the stratosphere the CH$_4$ oxidation contribution leads to a $\sim$0.5-1.5% dec$^{-1}$ increase in H$_2$O, the larger of those trends tending to be throughout the SH and above $\sim$30 km in the NH.

In each altitude/latitude bin, the CH$_4$ oxidation contribution was determined using the lagged $\frac{d[CH_4]_{entry}}{dt}$ value that corresponds to that bin's lag time determined for $T_{lag}$ (Fig. 7). The CH$_4$ oxidation contribution was then subtracted from that bin's residual H$_2$O trend that used AO, SAO, F10.7 cm, ACE-FTS N$_2$O, and $T_{lag}$ time series as regressors. The final trend results for this method are shown in Fig. 12, and it can be seen from the middle panel that when accounting for the non-linear increase in CH$_4$ there are more regions of the stratosphere where there are significant residual H$_2$O trends (than when assuming a constant increase). In the roughly 30-70°S, 20-35 km region, there remains a significant residual H$_2$O trend of 1.0-2.5% dec$^{-1}$. The residual trend is smaller in the same altitude/latitude region in the NH, between 0.9 and 1.7% dec$^{-1}$, although near 60°N

the region of significant trend extends from up to 55 km. There is also a significant increase of ~1% dec$^{-1}$ in parts of the SH low-latitude region above 45 km. These results indicate that there is at least one additional source of increasing $H_2O$ in multiple regions within the stratosphere that has not been accounted for. It is likely that the regions with unaccounted for trends are actually larger because, as discussed in Sect. 3 the lag times used to determine $\frac{d[CH_4]_{entry}}{dt}$ values only account for transit times from the entry point and does not account for mixing or differences in transit pathways (Poshyvailo-Strube et al., 2022). The effects on the regression results due to any correlation between the AO components and $T_{lag}$ or $N_2O$ are discussed in Appendix A2.

As previously mentioned, Wrotny et al. (2010) determined that measurements are consistent with $\alpha$ having a value of up to 3.7 that could account for $H_2O$ production via oxidation of $H_2$. Therefore, the $H_2O$ trend calculation was done again using the same regressors (AO, SAO, $N_2O$, $T_{lag}$, F10.7 cm) and the time-lagged $CH_4$ entry trends, but using a constant value of $\alpha = 3.7$ at all altitudes and latitudes. The results of the residual $H_2O$ trends are shown in Fig. 13 and are not statistically significant in nearly every bin. It should be noted that the value of $\alpha$ is not being changed in order to "optimize" the results, an extreme acceptable value was used simply to determine what effect that value would have on the calculated $H_2O$ trends. Although it is unlikely that the maximum value of $\alpha = 3.7$ is appropriate for all altitudes and latitudes, these results indicate that this higher value of $\alpha$ could be consistent with the calculated ACE-FTS $H_2O$ trends, especially in the mid-stratospheric extra-tropics, with the additional production due to increasing tropical tropospheric $H_2$ concentrations. In order to inform further analysis, a model study should be conducted investigating how $H_2$ concentrations have been changing over the course of the ACE-FTS mission lifetime.

One other source of stratospheric $H_2O$ that has not been accounted for is convective moistening. In the troposphere, deep convection systems can transport ice particles into the tropopause region and overshooting cloud tops can directly inject water vapour and ice into the lower stratosphere. Recent model studies (e.g., Dauhut and Hohenegger, 2022; Ueyama et al., 2023) have estimated that convective moistening contributes ~10% of the lower stratospheric $H_2O$ budget, and can contribute up to ~45% in monsoon regions (Dessler and Sherwood, 2004; Hanisco et al., 2007; Tinney and Homeyer, 2021). Ueyama et al. (2023) estimated the global inter-annual variation of lower stratospheric $H_2O$ produced via deep convection between 2006 and 2016 to be on the order of a few percent (0.05-0.1 ppmv), however the time period was too short to determine any significant trend. Further investigation is needed in order to determine if any longer-term changes in convection are influencing changes in stratospheric $H_2O$.

## 5 Conclusions

Measurements from ACE-FTS show that between 2004 and 2022 $H_2O$ concentrations have significantly increased at a rate of approximately 1-5% dec$^{-1}$ throughout nearly all of the stratosphere. This study uses ACE-FTS measurements of $H_2O$, $CH_4$, and $N_2O$, along with CMAM tropical upper-tropospheric $CH_4$ and ERA5 reanalysis tropical tropopause temperatures, to quantify the relative contributions of different sources of these $H_2O$ increases. The main sources are,

– increasing tropical tropopause region temperatures. This is the main source of increasing $H_2O$ in the tropical lower stratosphere. It accounts for $H_2O$ increases of,

         – $\sim$2-4% dec$^{-1}$ between 17 and 23 km in the tropics,

         – $\sim$1-2% dec$^{-1}$ between 23 and 50 km in the tropics, and

         – $\sim$1-2% dec$^{-1}$ 17-19 km in the mid-latitudes.

– structural BDC changes, which lead to,

         – $H_2O$ increases of $\sim$1-2% dec$^{-1}$ in NH mid-latitudes near 20-30 km, and

         – $H_2O$ decreases of $\sim$1-2% dec$^{-1}$ in SH mid-latitudes near 25-30 km and in NH mid-latitudes near 33-43 km.

     – Increasing $CH_4$ oxidation, which causes increases in $H_2O$ on the order of,

         – $\sim$1-2% dec$^{-1}$ above $\sim$30 km at all latitudes and above $\sim$20 km in SH.

The solar influence on stratospheric $H_2O$ was also investigated by regressing to F10.7 cm solar flux indicies. Its contribution to the stratospheric $H_2O$ trends was less than 0.5% dec$^{-1}$ in all altitude/latitude bins.

   These sources combined account for all significant stratospheric $H_2O$ trends except for a remaining $\sim$1-2% dec$^{-1}$ increase around 30-70° latitude in both hemispheres in the mid-stratosphere ($\sim$20-35 km). These remaining trends can be accounted for by substituting the altitude-dependent $CH_4$ oxidation $H_2O$ yield for a constant value of $\alpha = 3.7$ (upper limit from Wrotny

et al. (2010)), possibly indicating that these increases may be due to increasing concentrations of $H_2$, which also oxidizes to produce $H_2O$.

   Yet, it remains that the measured stratospheric $H_2O$ trends currently cannot be fully explained. As time goes on and more and more satellite limb sounding missions are coming to an end, for the sake of continuity it is vital that these types of atmospheric trends are fully understood—especially if there are going to be temporal gaps between the operational periods of current and

335 future instruments. There is an urgent need for new satellite missions to continue this observational record to enable more reliable trend estimation, as are model studies to determine what influence changes in processes such as $H_2$ oxidation and deep convection—and other possible sources—are having on the stratospheric regions where the full $H_2O$ trends cannot be fully accounted for.

*Code availability.* to be provided

*Data availability.* The ACE-FTS Level 2 data can be obtained via the ACE database (registration required), https://databace.scisat.ca/level2/ (ACE-FTS, 2024). The ACE-FTS data quality flags used for filtering the dataset can be accessed at https://doi.org/10.5683/SP2/BC4ATC (Sheese and Walker, 2024). CMAM39-SD data were obtained from ftp://crd-data-donnees-rdc.ec.gc.ca/pub/CCCMA/dplummer/CMAM39-SD_6hr. ERA5 data was obtained through the Copernicus Climate Change Service at https://cds.climate.copernicus.eu.

*Author contributions.* PES performed the analysis and wrote the manuscript. KAW led the project, gave insight to the ACE-FTS data, and helped edit the manuscript. CDB led the ACE-FTS retrievals, provided insight into the ACE-FTS data. DAP did the model experiments in CMAM39 and gave insight to the results. All authors contributed to the final version of the manuscript.

*Competing interests.* The authors have no competing interests to declare

*Acknowledgements.* This project was funded by the Canadian Space Agency (CSA). The Atmospheric Chemistry Experiment is a Canadian-led mission mainly supported by the CSA. We acknowledge Peter Bernath, who is the PI of the ACE mission. The development of the CMAM39 data set was funded by the CSA. We thank Ted Shepherd, Dylan Jones, and John Scinocca for their leadership and support of the CMAM39 Project.

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

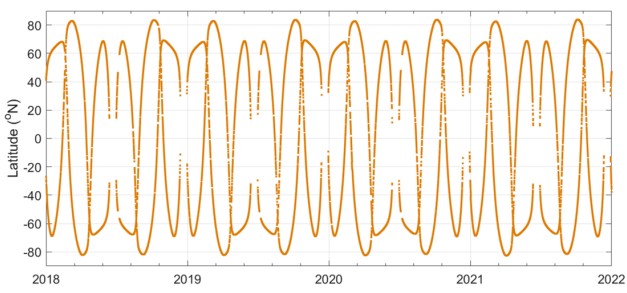

**Figure 1.** Latitudinal coverage of the ACE-FTS measurements for 2018-2022.

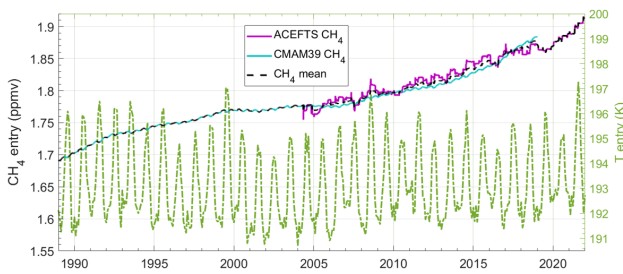

**Figure 2.** Mean CH$_4$ time series for 15°S-15°N and 100-200 hPa. ACE-FTS data (cyan) are 180-day running means and the CMAM39 data (magenta) are daily means. The mean of ACE-FTS and CMAM39 (black dashes) was used to determine the CH$_4$ entry trend. Also, ERA5 temperatures (green dot-dash) are monthly mean values for 15°S-15°N at 100 hPa.

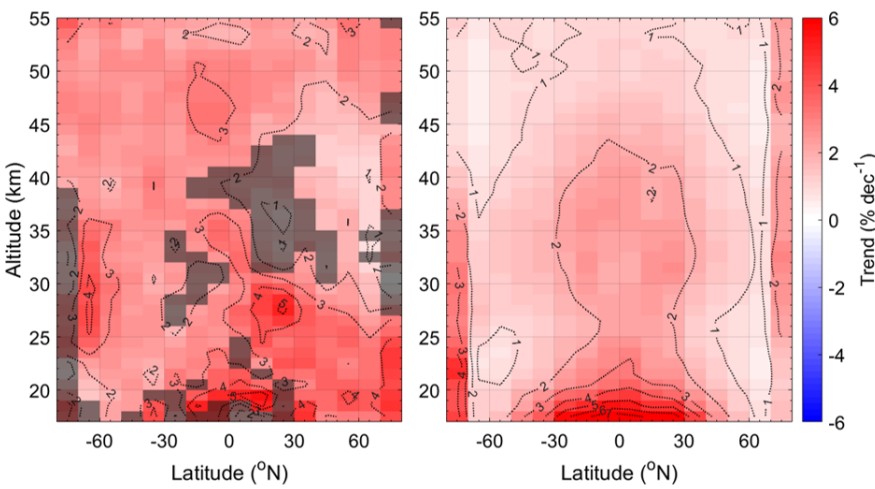

**Figure 3.** ACE-FTS H$_2$O trends. (left) Full trends (where only regressing to semi-annual and annual cycles) and (right) corresponding trend uncertainty to a confidence level of 95%. Shaded regions in left panel indicate regions with no significant trend to within 95%.

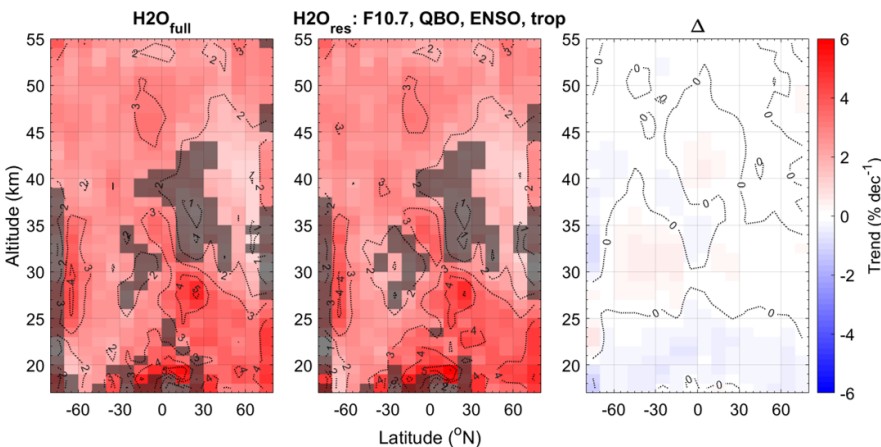

**Figure 4.** ACE-FTS $H_2O$ trends. (left) Full trends (where only regressing to semi-annual and annual cycles). (middle) Residual $H_2O$ trends after regressing to F10.7 cm, QBO, ENSO, and tropopause pressure time series and semi-annual and annual cycles. (right) The difference between the full trends and the residual trends. Shaded regions in left and middle panels indicate regions with no significant trend to within 95%.

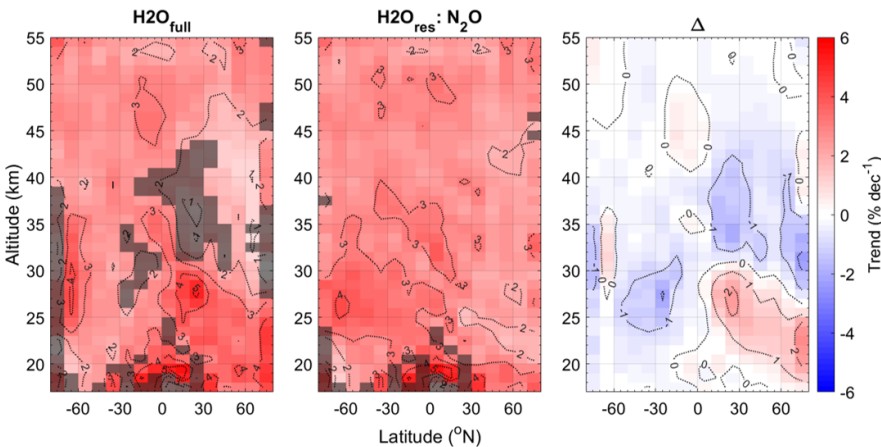

**Figure 5.** Same as Fig. 4, except the residual H$_2$O trends (middle) are after regressing to N$_2$O and semi-annual and annual cycles.

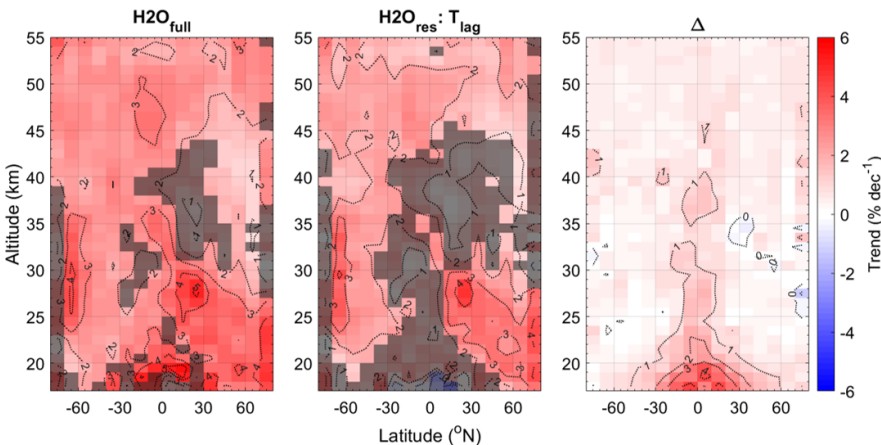

**Figure 6.** Same as Fig. 4, except the residual H₂O trends (middle) are after regressing to ERA5 temperatures with empirically determined time lags ($T_{lag}$) and semi-annual and annual cycles.

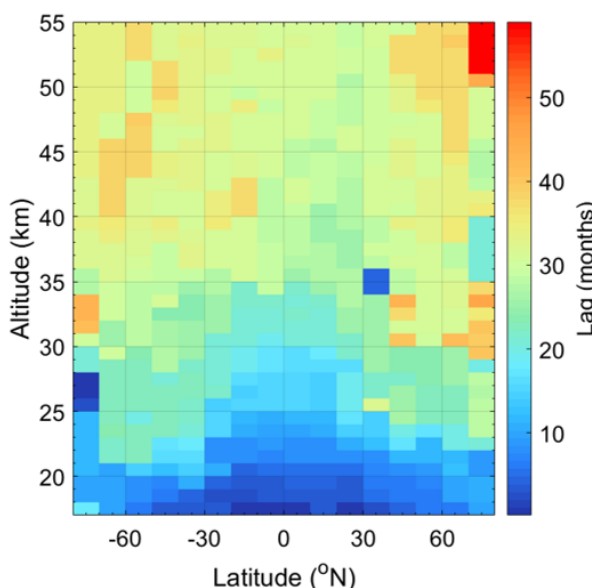

**Figure 7.** Lag times introduced to the ERA5 temperature time series ($T_{lag}$) that minimize the residuals between the ACE-FTS data and the MLR fit.

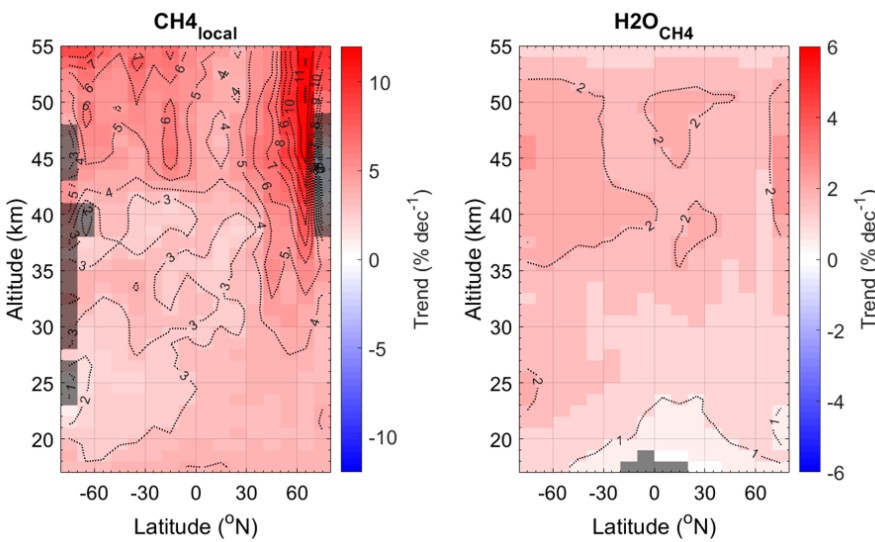

**Figure 8.** (left) ACE-FTS $CH_4$ local trends and (right) contribution of $CH_4$ oxidation to ACE-FTS $H_2O$ trends. Shaded regions indicate regions with no significant trend to within 95%.

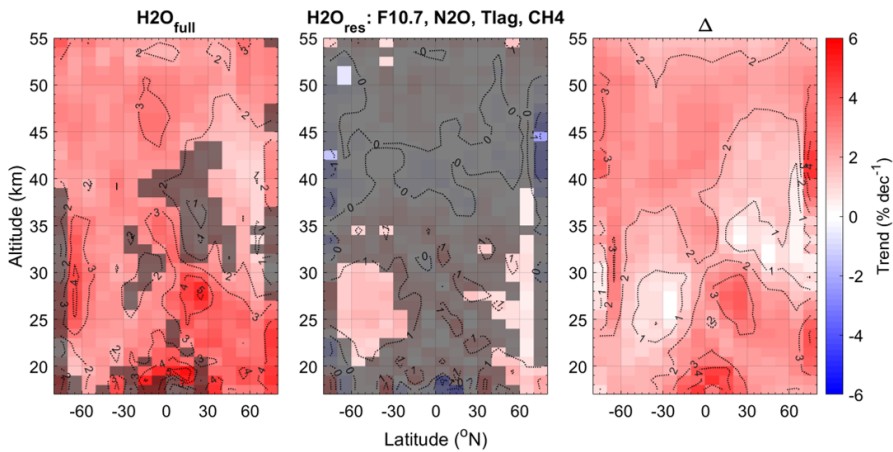

**Figure 9.** Same as Fig. 4, except the residual H$_2$O trends (middle) are after regressing to semi-annual and annual cycles, F10.7 cm flux, N$_2$O, $T_{lag}$ time series and accounting for CH$_4$ oxidation.

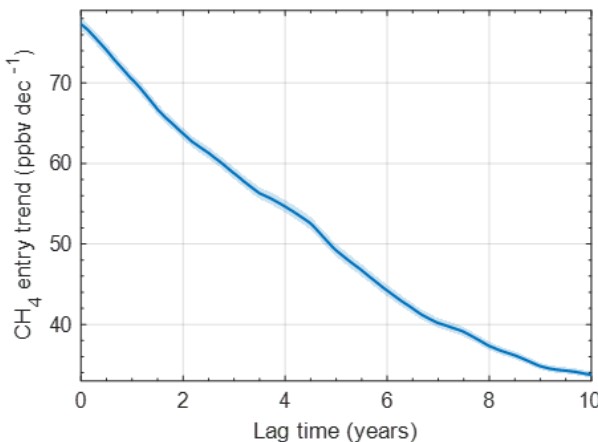

**Figure 10.** 18-year CH₄ entry trends, as a function of lag time, for time periods of 1994-2012 (lag of 10 years) to 2004-2022 (lag of 0 years). Shaded region represents the 95% confidence levels.

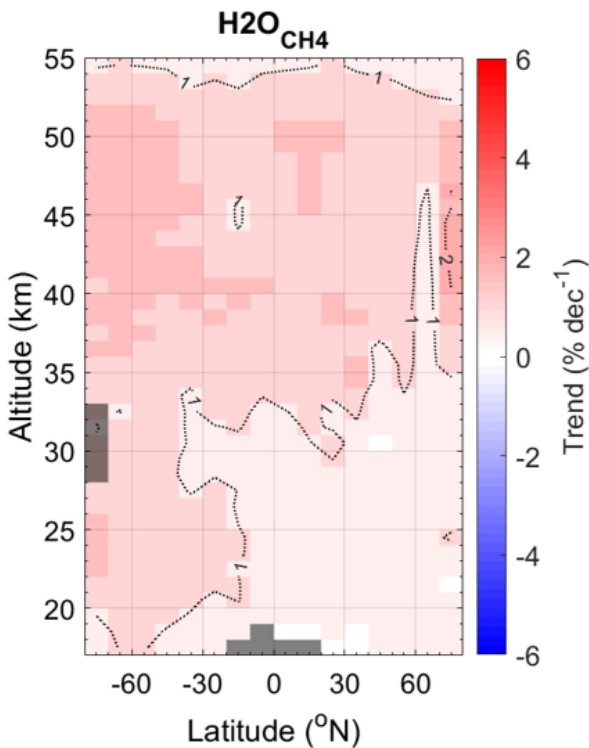

**Figure 11.** Trends in $H_2O$ due to $CH_4$ oxidation using time-lagged $CH_4$ entry trend values (based on $T_{lag}$ lag times).

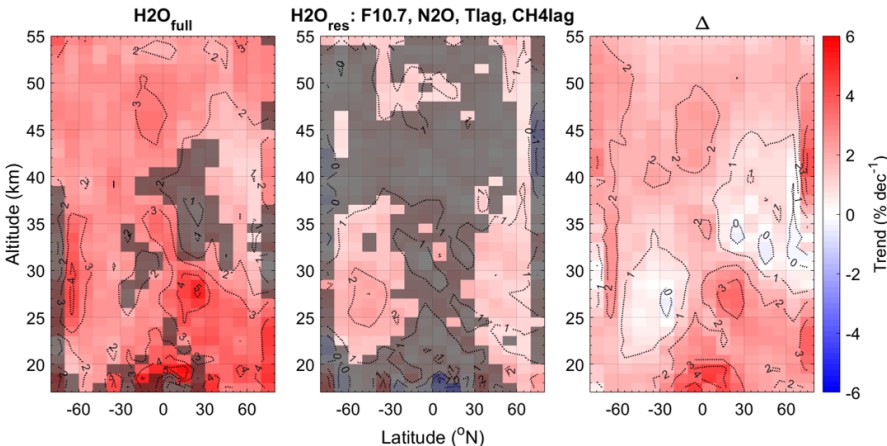

**Figure 12.** Same as Fig. 9 except time-lagged CH$_4$ entry trend values (based on $T_{lag}$ lag times) were used when accounting for CH$_4$ oxidation.

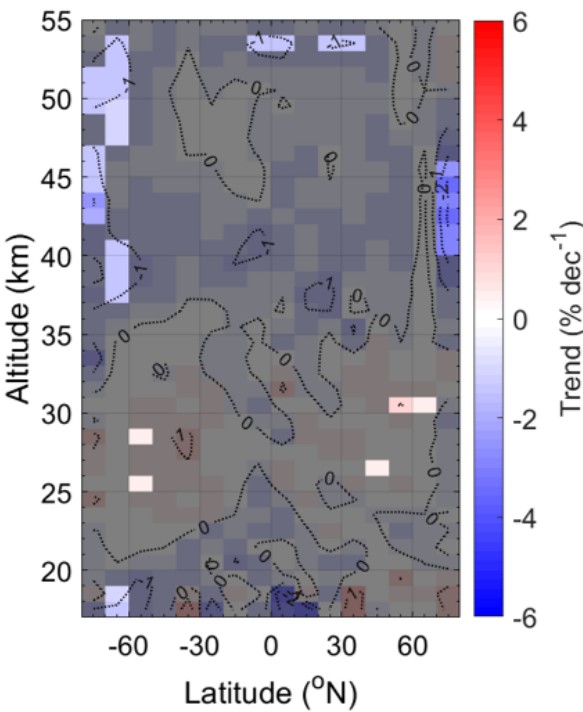

**Figure 13.** Residual $H_2O$ trends after regressing to semi-annual and annual cycles, F10.7 cm flux, $N_2O$, $T_{lag}$ time series and accounting for $CH_4$ oxidation using time-lagged $CH_4$ entry values and a $H_2O$ yield value of $\alpha = 3.7$ in all altitude/latitude bins. Shaded regions indicate regions with no significant trend to within 95%.

**Appendix A**

**A1    Time lagged ENSO and tropopause pressure**

The effects of ENSO conditions and tropopause pressure on stratospheric $H_2O$, however, are not instantaneous. Similar to the time-lagging method described in Sect. 3, the ENSO and trop time series were lagged to find lag times in each bin that maximize the correlations between local $H_2O$ and the respective index time series. The influence of lagged ENSO and lagged trop on $H_2O$ trends are shown in the top panel of Fig. A1 and the respective corresponding optimal lag times in months are shown on the bottom. The influence of both lagged ENSO and lagged trop is still typically less than 1% dec$^{-1}$ throughout the

stratosphere.

**A2    Correlation with AO**

When accounting for $CH_4$ oxidation with time-lagged entry trends and regressing to $T_{lag}$ and $N_2O$ and SAO and no AO components, the resulting residual trends are not meaningfully different below $\sim$42 km from the same case with AO as a regressor, as seen in Fig. A2. The main differences are above 42 km. Without including AO, the region with significant residual

trends in the low-mid latitudes is larger, but still on the order of 1% dec$-1$, the SH high latitudes exhibit negative residual trends on the order of -1% dec$^{-1}$, and the NH mid-high latitudes exhibit no significant trends. Therefore, further study is still needed to fully parse the $H_2O$ trends in the upper stratosphere.

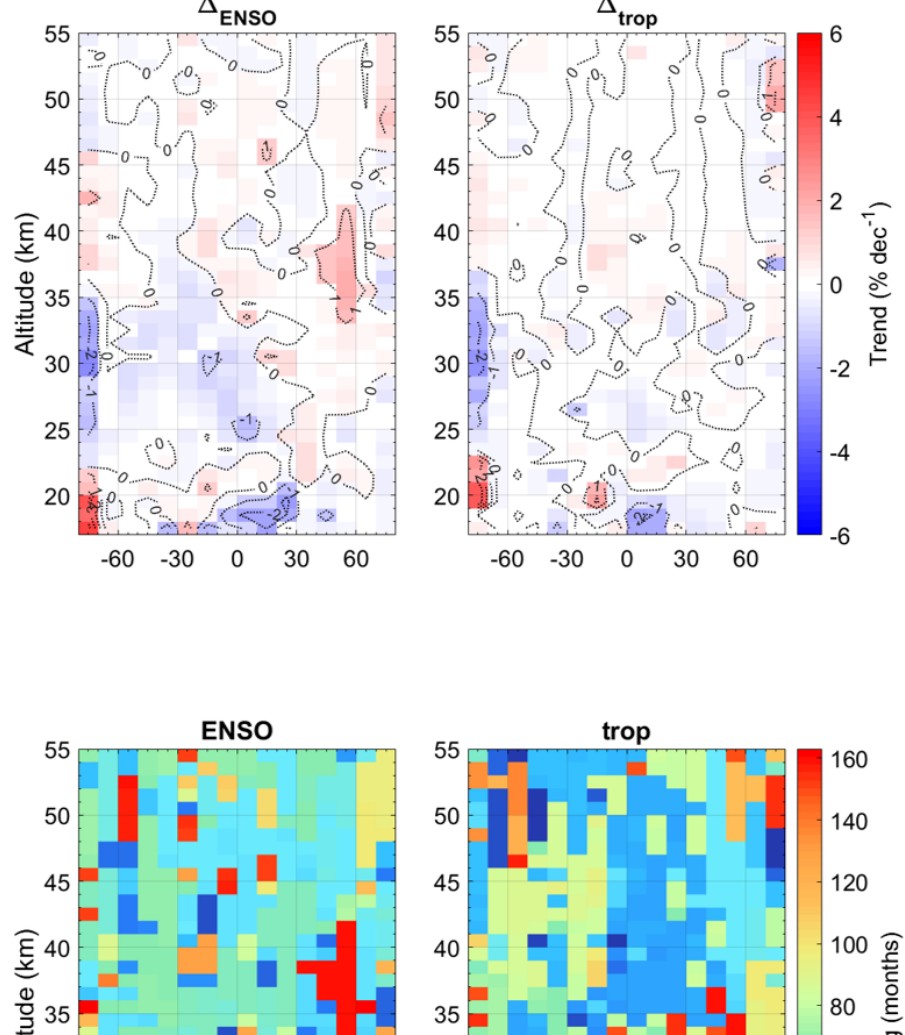

**Figure A1.** (top) The difference between the full trends and the residual trends when regressing to the lagged ENSO (left) and trop (right) time series and semi-annual and annual cycles. (bottom) The corresponding optimal lag times.

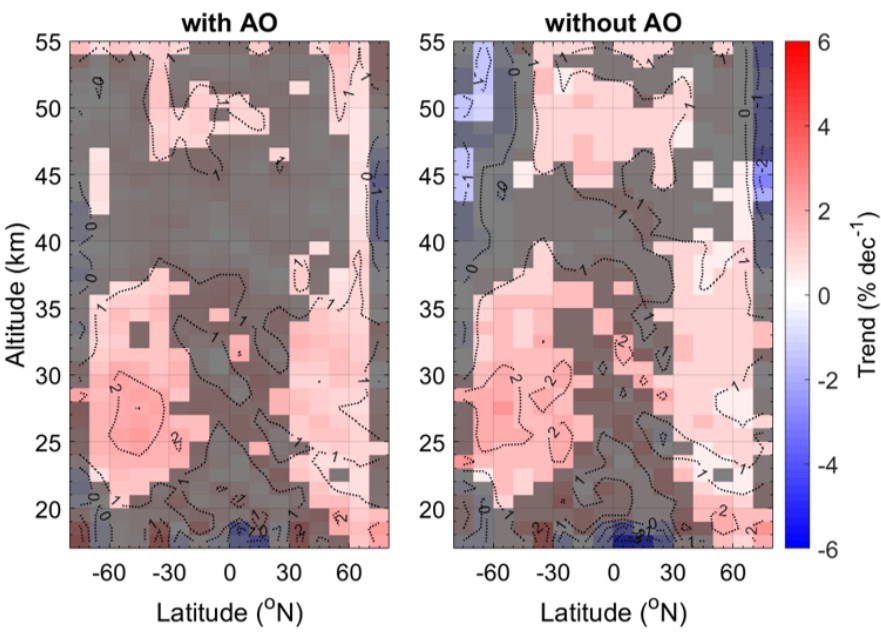

**Figure A2.** (left) Residual $H_2O$ trends after regressing to semi-annual and annual cycles, F10.7 cm flux, $N_2O$, $T_{lag}$ time series and accounting for $CH_4$ oxidation using time-lagged $CH_4$ entry values. (right) Same as (left) but without regressing to annual cycles. Shaded regions indicate regions with no significant trend to within 95%.