# Peer review of "Quantifying the sources of increasing stratospheric water vapour concentrations"

_EGUsphere, 2024_

## Author Comment (AC1)

**Response to Reviewers – "Quantifiying the sources of increasing stratospheric water vapour concentrations" by P. E. Sheese et al. (egusphere-2024-2946)**

*We thank the reviewers for their time and insight, and we appreciate their thoughtful comments. The reviewers' comments are given below with our responses in green italics.*

Reviewer 1:

Major Comments

1. The Title:

   Since the study covers the period 2004–2021, referring to the "21st century" in the title may be misleading. I suggest rephrasing the title to reflect the time span analyzed more accurately.

   *"in the 21ˢᵗ century" has been removed from the title*

2. Lines 3 & 49: 'None of them parse the trend'

   This statement is not entirely accurate. Yu et al. (2022) quantified the contributions of TTL temperature and methane oxidation to stratospheric water vapor trends. Please revise this part accordingly to acknowledge previous relevant studies.

   *It is true that there are studies that parse one or two sources at certain latitudes. We are the first to parse trends due to the three main sources at all latitudes. We have now made this clearer in the text.*

   *The abstract now reads, "…but none have simultaneously quantified the contributions from all main sources (temperature variations in the tropical tropopause region, changes in the Brewer-Dobson circulation, and changes in methane ($CH_4$) concentrations and oxidation) at all latitudes."*

   *The introduction now states, "…none of them parse trends in order to determine the relative contributions from each of the three main sources throughout the stratosphere."*

3. Contribution of Brewer-Dobson Circulation and Tropical Tropopause Temperature:

   This section needs further clarity, as the results are somewhat confusing.

   - I recommend adding time series of tropical tropopause temperature and Brewer-Dobson circulation, similar to what is presented in Figure 1, to provide additional context.

*The ERA5 temperatures are now also plotted in Fig 2 (previously Fig. 1). There is no BDC time series, as detailed in the paper, we use simultaneously measured $N_2O$ time series, which are different for each analyzed bin.*

o   Studies like Fu et al. (2019) and Polvani et al. (2018) report a slowdown of the Brewer-Dobson circulation during the first two decades of the 21st century. However, in your study, the Brewer-Dobson circulation is shown to increase stratospheric water vapor in certain regions (e.g., Southern Hemisphere high latitudes, Northern Hemisphere mid-latitudes) and decrease it elsewhere. Could you provide further explanation or discussion to reconcile this difference?

*We have added to Section 4.1:*

*"Climate models have long indicated that increasing concentrations of greenhouse gases in the lower atmosphere should lead to an acceleration of both the shallow and deep branches of the BDC (e.g., Butchart et al., 2014, and references therein). Multiple studies have examined measurements of atmospheric proxies for BDC changes and have detected accelerations in the shallow branch (below ~20 km), although not at higher altitudes in the deep branch (e.g., Engel et al., 2009; Diallo et al., 2012; Engel et al., 2017). However, recent studies have suggested that decreasing concentrations of ozone depleting substances in the stratosphere can lead to a deceleration of the BDC (e.g., Polvani et al., 2018; Fu et al., 2019). Polvani et al. (2018) analyzed data from WACCM runs for 1965-2080 and showed that between 2000 and 2080, the BDC is expected to slow down due to the removal of ozone depleting substances. Fu et al. (2019) show that within 2000-2018 there has been a slowing down of the BDC in the SH lower stratosphere (10-50 hPa), but no clear overall trend in the NH. This agrees with ACE-FTS showing a decrease in $H_2O$ due to circulation changes throughout most of the SH and a combination of increasing and decreasing circulation in the NH depending on the season. Since Fu et al., 2019 averaged data over the entire NH from 10-50 hPa, it is unknown whether the spatial patterns of the seasonal differences they reported would be in agreement with the BDC results in this study."*

o   Separating the contributions of the Brewer-Dobson circulation and tropopause temperature is challenging, as a weaker Brewer-Dobson circulation can lead to lower tropopause temperatures. I suggest conducting a more careful analysis to distinguish the respective influences of these two factors.

*Whatever influence the BDC has on tropopause temperatures, that influence is embedded in the temperature time series and that will be captured in the regression. We do not feel it is necessary to parse the $H_2O$ trends due to temperature changes further to separate those $H_2O$ trends due to temperature*

*changes due to BDC changes and those $H_2O$ trends due to temperature changes due to other influences.*

4.  Sampling and Reliability of Water Vapor Trends:

    The paper mentions that the ACE-FTS instrument has low sampling in the tropics, and that CMAM is used to estimate $CH_4$ trends. I recommend addressing the following points:

    o   In Section 2, please provide more detailed descriptions of the $H_2O$ sampling over different latitude bands.

        *A figure of the ACE-FTS latitudinal coverage is now included in Fig. 1 and is briefly discussed in the end of Sect. 2, "The latitudinal coverage of the instrument is shown in Fig.1. As can be seen, the annually-repeating measurements are made predominantly in the high latitudes, with the tropics being sampled roughly every three months. However, significant trends are still capable of being detected at the lower latitudes due to the long lifetime of the ACE-FTS data sets."*

    o   It would also be helpful to include an analysis of how reliable the tropical water vapor trends are, given the known sampling limitations.

        *We now explicitly show a plot of the water vapour trend uncertainties (including in the tropics) along side the water vapour trends in a new figure (Fig. 3), and we now explicitly state that the uncertainties are to 95% confidence level.*

5.  Line 114: Tropopause Height and Seasonality

    Since the height of the tropopause varies with season, have you accounted for seasonal variability? Additionally, have you tested the results using higher altitudes to ensure robustness?

    *The AO and SAO regressors are included in order to handle seasonal variability. At higher altitudes, the $H_2O$ trends due to tropical tropopause region temperatures and due to $CH_4$ oxidation are consistent, however $N_2O$ can no longer be used as an effective dynamical tracer as there is a lower thermospheric source of $N_2O$. We now also include a tropopause pressure index time series (briefly described in the methodology section) as a regressor for the results in Fig. 4.*

Specific Comments

1. Line 17:

   There is a typo in "greenhouse gases."

   *Both "gases" and "gasses" are acceptable plural forms, but we have changed it to the more standard spelling here and all other instances*

2. Line 82:

   The SWOOSH database (Davis et al., 2016) also utilizes ACE-FTS $H_2O$ data.

   *This reference is now included*

3. Line 219:

   The statement, "with an increase of less than 1% per decade near the tropical tropopause region," raises a question. Methane does not undergo oxidation in this region, so how can its contribution be non-zero?

   *This now states, "...with insignificant influence closer to the tropical tropopause region where there is expected to be little to no contribution from methane oxidation."*

4. Line 253 (Suggestion):

   As ACE-FTS provides HDO data, you might consider using it as a proxy for deep convection, following Hanisco et al. (2007). While this analysis may be beyond the scope of the current paper and the results may be uncertain, it could be worthwhile to explore how HDO could provide insights into the role of deep convection in the observed trends.

   *This is a fun suggestion! We will certainly look into that, but yes, it's undoubtedly outside the scope of this study.*

Reviewer 2

There are two main reasons I recommend a major revision of the paper:

First, although the findings related to stratospheric moistening after 2000, its hemispheric asymmetry associated with changes in the Brewer-Dobson circulation, the tropical cold point temperature, and CH4 oxidation are interesting, they are not entirely novel. These phenomena have already been discussed in previous studies, including Konopka et al., 2022, and Tao et al., 2023.

*We have clarified our point in the text that this study is the first to quantify $H_2O$ trends due to all three sources (tropopause temperatures, BDC, and $CH_4$) at once as a function of latitude and altitude.*

*The abstract now reads, "…but none have simultaneously quantified the contributions from all main sources (temperature variations in the tropical tropopause region, changes in the Brewer-Dobson circulation, and changes in methane ($CH_4$) concentrations and oxidation) at all latitudes."*

*The introduction now states, "…none of them parse trends in order to determine the relative contributions from each of the three main sources throughout the stratosphere."*

Konopka et al., 2022, doi.org/10.1029/2021GL097609

*- this study discusses one source but does not discuss methane or general circulation changes*

Tao et al., 2023, doi.org/10.1038/s43247-023-01094-9

*-this study does discuss all three sources, but does not explicitly quantify their contributions to the overall trends*

Second, there is a need to explain more critically all the additional assumptions beyond the multiple linear regression model that were used to conclude the potential importance of hydrogen trends (see below).

Minor points:

Title: The phrase "in the 21st century" may be misleading, as only 25 years of data are available. Please consider rephrasing.

*The title no longer states "in the 21$^{st}$ century"*

Abstract: The sentence "Previous studies have estimated stratospheric H2O trends, but none have simultaneously quantified..." is inaccurate (see the point above on existing studies).

*The abstract now reads, "...but none have simultaneously quantified the contributions from all main sources (temperature variations in the tropical tropopause region, changes in the Brewer-Dobson circulation, and changes in methane ($CH_4$) concentrations and oxidation) at all latitudes."*

Abstract: The phrase "these unaccounted... sources could potentially" may need rephrasing for clarity.

*This now reads, "Results indicate that these unaccounted for increases could potentially be explained by increases in upper tropospheric molecular hydrogen."*

Methodology: The multi-linear regression model is not well explained. For example, the annual oscillation term appears to mix harmonics and regressors. Additionally, there is no explanation of the time lags, which are introduced later to improve the analysis.

*We have added some more detail about the independence of the different regressors with each other. We now state near the end of Sect. 3, "At all altitudes and latitudes, the absolute correlation between measured $N_2O$ and Tlag time series is less than 0.35 and is typically below 0.2. The same is true for the correlation between $N_2O$ and the seasonal cycles, and for Tlag and SAO—however between Tlag and SAO the correlation is typically on the order of 0.2-0.3. Tlag and AO, however, are not independent as the tropopause region temperatures exhibit a significant annual oscillation. The implications of this are discussed in Sect. 4.2."*

*In the results (Sect 4.2) we now include, "When accounting for $CH_4$ oxidation with time-lagged entry trends and regressing to Tlag and $N_2O$ and SAO and no AO components, the resulting residual trends are not meaningfully different below ~42 km from the same case with AO as a regressor, as seen in Fig. 14. The main differences are above 42 km. Without including AO, the region with significant residual trends in the low-mid latitudes is larger, but still on the order of 1% $dec^{-1}$, the SH high latitudes exhibit negative residual trends on the order of -1% $dec^{-1}$, and the NH mid-high latitudes exhibit no significant trends. Therefore, further study is still needed to fully parse the $H_2O$ trends in the upper stratosphere."*

*We respectfully disagree that "there is no explanation of the time lags, which are introduced later to improve the analysis." Lines 115-124 (of the original manuscript), within the methodology section, discusses this in specific detail. The text details how the time series was lagged as well as the physical rationale behind the lagging.*

L225-234: This section represents the most critical part of the manuscript, particularly in its attempt to extend the definition of d[CH4]entry/dt by incorporating time lags. This approach differs from that in Fig. 5, where time lags are defined by minimizing the residuals between the data and the multi-linear regression (MLR) model. In this section, however, the same time lags are applied to redefine d[CH4]entry/dt, making their use appear somewhat "a posteriori." It is unclear why the authors did not apply the well-established formalism for analyzing stratospheric trends introduced in Hegglin et al., 2014, and subsequently applied in many studies, such as

Poshyvailo-Strube et al., 2022 (doi.org/10.5194/acp-22-9895-2022) and Tao et al., 2023 (see reference above). Additionally, the term "alpha" is defined differently in this paper compared to previous references, which may lead to confusion.

*Because we are not regressing to $CH_4$ as we are with temperature, a different approach is taken in the method for determining lag times for $CH_4$ entry. The text in Sect. 4.2 now reads,*

*"This method of using lagged $CH_4$ entry trends was employed by Hegglin et al. (2014). That study used time-lagged $CH_4$ entry trends with lag times corresponding to the local mean age-of-air. However, ACE-FTS does not have validated measurements of age-of-air throughout the stratosphere, and therefore the optimal lag times from the temperature regression (Fig. 8) are used as a proxy for mean age-of-air."*

L245-250: The complexity further increases when discussing variable values of alpha. The main objective of multi-linear regression is to minimize the residual by simultaneously varying all relevant parameters, including the lag times and the alpha parameter. A critical discussion justifying the "simplified procedure" used in this paper is necessary.

*As we have clearly detailed in the text, we are not regressing to $CH_4$, and therefore there is no way of minimizing residuals by varying alpha or lag times. The only thing that can be done with alpha is use different values and comment on the residual $H_2O$ trends. For the bulk of the study, we have used the recommended values. We now clarify near the end of Sect. 4.2, "It should be noted that the value of α is not being changed in order to "optimize" the results, an extreme acceptable value was used simply to determine what effect that value would have on the calculated $H_2O$ trends."*

Reviewer 3

Minor comment (Description and discussion of the methodology, Sect. 3):

I have one general and a few more specific comments here. My general comment concerns the choice of regressors in the MLR method: To what degree are the chosen regressors independent? In particular there are interrelationships between tropical tropopause temperatures and stratospheric circulation (here N2O is chosen as proxy for these), as well as between these two and annual cycle, QBO, ENSO, etc. I know that such a regression approach is applied frequently and I don't argue for changing the methodology. But I think it would be worth discussing these aspects (and the ones below) in more detail, either in the method section or later in a discussion section.

*As mentioned above, we have added some more detail about the independence of the different regressors with each other. As noted in the response to Reviewer 2, we now state near the end of Sect.3, "At all altitudes and latitudes, the absolute correlation between measured $N_2O$ and Tlag time series is less than 0.35 and is typically below 0.2. The same is true for the correlation between $N_2O$ and the seasonal cycles, and for Tlag and SAO—however between Tlag and SAO the correlation is typically on the order of 0.2-0.3. Tlag and AO, however, are not independent as the tropopause region temperatures exhibit a significant annual oscillation. The implications of this are discussed in Sect. 4.2."*

*In the results (Sect. 4.2) we now include (as noted in response to Reviewer 2), "When accounting for $CH_4$ oxidation with time-lagged entry trends and regressing to Tlag and $N_2O$ and SAO and no AO components, the resulting residual trends are not meaningfully different below ~42 km from the same case with AO as a regressor, as seen in Fig. 14. The main differences are above 42 km. Without including AO, the region with significant residual trends in the low-mid latitudes is larger, but still on the order of 1% dec$^{-1}$, the SH high latitudes exhibit negative residual trends on the order of -1% dec$^{-1}$, and the NH mid-high latitudes exhibit no significant trends. Therefore, further study is still needed to fully parse the $H_2O$ trends in the upper stratosphere."*

Another issue which could be discussed in more detail is the effect of ENSO. Aren't ENSO effects on stratospheric water vapor likely lagged (compared to the ENSO index used in the MLR method), in particular when considering upper stratospheric levels and high latitude regions. Hence including ENSO in the regression could likely be improved by taking into account lag time, as is done already for the impact of tropopause temperatures. Again, I'm not suggesting new analysis but only a more detailed discussion.

*We have tried lagging ENSO both with the same lag times as with the temperature lag times and with separate lag times that minimize residuals in the regression. In both cases, the differences in the $H_2O$ trends are typically less than a percent per decade and a new figure (Fig. 5) has been added in Sect. 4.1 to show how lagging ENSO and tropopause pressure indices affects the regression. We have added to the text in Sect. 4.1, "The effects of ENSO conditions and tropopause pressure on stratospheric $H_2O$, however, are not instantaneous. Similar to the time-lagging method described in Sect. 3, the ENSO and trop time series were lagged to find lag times in each bin that maximize the correlations between local $H_2O$ and the respective index time series. The influence of lagged ENSO and lagged trop on $H_2O$ trends are shown in the top panel of Fig. 5 and the respective corresponding optimal lag times in months are shown on the bottom. The influence of both lagged ENSO and lagged trop is still typically less than 1 % dec$^{-1}$ throughout the stratosphere."*

Also concerning the calculation of the methane oxidation effect there is an approximation applied but not clearly explained. The exact calculation of the CH4-entry values in Eq. 2 would require the convolution with the stratospheric age spectrum. As this is not available, here a lag time is used as approximation. A recent paper discusses the effect of approximating the propagation of methane entry mixing ratios in a different context (Poshyvailo-Strube et al., 2022, https://doi.org/10.5194/acp-22-9895-2022). I'd suggest to discuss also this approximation in a bit more detail.

*In the methodology section (Sect. 3), we now state, "...it was found that the maximum lag time required to minimize the residuals in any bin was 60 months, whereas stratospheric mean age-of-air estimates tend to be on the order of 0-15 years, depending on altitude and latitude. This can be explained by the fact that lagging the temperature time series assumes that the measured air parcel is a singular parcel that traveled from the entry point to the measurement location with a particular transit time and does not account for mixing or changes in transit pathways (Poshyvailo-Strube et al., 2022)."*

*In Sect. 4.2 we now state, "It is likely that the regions with unaccounted for trends are actually larger because, as discussed in Sect. 4.2 the lag times used to determine dCH4entry/dt values only account for transit times from the entry point and does not account for mixing or differences in transit pathways (Poshyvailo-Strube et al., 2022)."*

A more technical comment concerns the notation in Eq. 1: I find the indices i, j here somewhat confusing. In particular, shouldn't there also be a sum over j? Later in L94, aren't the cos and sin functions the r_i's in Eq. 1, so shouldn't there be a similar index i in these and the beta-coefficients? In Eq. 4 the j-index is just 2 for AO and SAO, is this correct? Please explain the notation more clearly.

*Instead of discussing this in terms of j harmonics, the equation has been simplified to, $y_{fit} = \beta_0 + \beta_1 l(t) + \sum_i \beta_i r_i(t)$, and the AO and SAO are now described as "-two annual oscillation terms cos2πt and sin2πt (AO) and two semi-annual oscillation terms cos 4πt and sin 4πt(SAO), with t measured in years;"*

Specific comments:

L10: I'd precise the formulation here: "increases in the Northern Hemisphere below about 30km and decreases above and in the Southern Hemisphere throughout the stratosphere"

*We agree that the more precise language is ideal, however we are already at the abstract word count limit and cannot cut information from anywhere else in the abstract.*

L30: Actually, the first study emphasizing the general point of stratospheric dehydration at the tropical tropopause would be Brewer (1949), the main point of the paper cited here is the impact of regional temperature anomalies. I leave the decision what to cite here to the authors...

*The Brewer paper has been added*

L79: isotopologues

*This has been corrected*

L114: How can the water vapor time series over 2004-2021 be regressed against a temperature time series which ends in 2018? Please clarify.

*"2018" was a mistake. This has been amended to "2022".*

L171: To better distinguish the circulation effect from the effect of tropopause temperature changes I'd suggest to write here "structural Brewer-Dobson circulation changes". Perhaps this would be a better formulation also at other places in the manuscript.

*We have changed this and other instances to "structural Brewer-Dobson circulation changes"*

L184: I was a bit confused about this sentence, as "residual trends" are shown only in the right panel of Fig. 2. I'd suggest to write either "right panel of Fig. 2" or just "Fig. 2".

*This section has changed due to adding a plot of full trend uncertainties, and it now reads, ", and these are shown to compare to the residual $H_2O$ trends when regressing to the F10.7 cm,*

*QBO, ENSO, and trop time series, shown in the middle panel." The right panel is the difference between the full trends and these residual trends.*

L192: Also here I don't find the formulation very clear. My suggestion: "The remaining trends are all increasing (Fig. 3, middle panel), but the hemispheric asymmetry in the lower stratosphere flips sign and remaining trends are more strongly positive in the SH (about 3%/dec) compared to the NH (1-3%/dec). The difference ... circulation, showing that the net hemispheric asymmetry in water vapor trends can be attributed to changes in stratospheric circulation. In particular, the circulation influence ... " These findings are consistent with results of another recent paper by Tao et al. (2023, https://doi.org/10.1038/s43247-023-01094-9) based on different data and I'd suggest to mention that here.

*This now reads, "The remaining trends are all increasing (Fig. 6, middle panel) but the hemispheric asymmetry in the lower stratosphere flips sign and the remaining trends are more strongly positive in the SH (~3% dec$^{-1}$) compared to the NH (1-3% dec$^{-1}$). Above ~35 km the remaining H$_2$O trends are more consistent between hemispheres, on the order of 1-3% dec$^{-1}$. These results are consistent with those of Tao et al. (2023). The differences (right panel of Fig. 6) represent the trend in H$_2$O due to changes in the general circulation, showing that the net hemispheric asymmetry in H$_2$O trends can be attributed to changes in stratospheric circulation. In particular, the influence can be to either increase or decrease local H$_2$O concentrations, depending on the region."*

L231: I think the correct reference here is "Fig. 6b".

*The figure reference has been corrected. It now refers to "Fig. 9, right panel."*

L273: To me the H2O increases at high latitudes and altitudes between 25-35km in the SH due to circulation changes in Fig. 3c seem very small and almost negligible. Thus, I'd not highlight them here in the conclusions.

*Agreed. This now simply reads, "-H$_2$O increases of ~1-2% dec$^{-1}$ in NH mid-latitudes near 20-30~km, and…"*

L287: Maybe worth to include a sentence after "...instruments." similar to: "There is urgent need for new satellite missions to continue the observational record for more reliable trend estimation, as are model studies to determine ..."

*This is a great suggestion and has been added. "There is an urgent need for new satellite missions to continue this observational record to enable more reliable trend estimation, as are model studies to determine what influence changes in processes…"*

---

## Author Response (AR2)

Response to Editor Report – Quantifying the sources of increasing stratospheric water vapour concentrations" by P. E. Sheese et al. (egusphere-2024-2946)

*We thank the reviewers for the time they've spent on improving the paper and for their thoughtful comments.*

*Below are the reviewers' comments with our responses in green italics.*

Reviewer 1

After this round of revision, I still find the motivation for this work unconvincing. As I mentioned in Comment 3, temperature and the Brewer-Dobson circulation (BDC) are inherently linked. My concern is not that the authors should further separate $H_2O$ trends due to temperature changes into different contributing factors (e.g., BDC-driven vs. other influences). Rather, my point is that temperature and BDC are not independent variables, then the multivariate regression will be less meaningful.
A key issue is that when a BDC time series is not available, the $N_2O$ time series should be included in the analysis to ensure it is not overly correlated with temperature before performing multivariate regression (maybe both should remove the climatology). Without addressing this, the regression results between BDC and water vapor could simply reflect an indirect relationship—where temperature influences both BDC and water vapor—rather than a direct effect of BDC on water vapor.

*The previous iteration of the manuscript and the response to the reviewers stated, "At all altitudes and latitudes, the absolute correlation between measured $N_2O$ and $T_{lag}$ time series is less than 0.35 and is typically below 0.2." We've now added a sentence directly before this, which states, "To ensure that it is appropriate to simultaneously use $T_{lag}$ and local $N_2O$ time series as regressors, the correlation between these time series was calculated in each altitude and latitude bin."*

Additionally, Dessler et al. (2013) identified QBO, BDC, and temperature at 500 hPa (rather than the tropopause region) as the primary factors influencing water vapor trends. This choice is reasonable because 500 hPa temperature is more independent of QBO and BDC.

*Dessler et al. (2013) make use of a temperature time series from 500 hPa, but they do not discuss or indicate that using a time series from 500 hPa is preferable over time series from any other pressure level. We are using a temperature time series from 100 hPa in order to remain consistent with the methodology of Hegglin et al. (2014; 10.1038/NGEO2236).*

To strengthen the study's justification, the authors should clarify why including both BDC and tropopause temperature is necessary and valid, given their interdependence. Without addressing this, the statement in the reply to Reviewers 1 & 2 that "this is the first paper to study all three factors" does not seem fully justified, which, could be why previous studies have not treated temperature, BDC, and methane (which is truly independent) as three separate factors influencing water vapor.

*Other studies (e.g. Hegglin et al. (2014), Tao et al. (2023)) have treated tropical tropopause region temperatures and local BDC changes independently. As written in the manuscript and clarified above, the lagged temperature time series and local $N_2O$ time series exhibit low correlation and can therefore be treated as independent variables within the multivariable regression. Further, we cannot justify the statement "this is the first paper to study all three factors," as we at no point stated or implied this. In our initial responses, to Reviewer 1 we wrote, "We are the first to parse trends due to the three main sources at all latitudes." To Reviewer 2 we wrote, "...this study is the first to quantify $H_2O$ trends due to all three sources (tropopause temperatures, BDC, and $CH_4$) at once as a function of latitude and altitude." Similarly, the abstract states, "...none have simultaneously quantified the contributions from all main sources (temperature variations in the tropical tropopause region, changes in the Brewer-Dobson circulation, and changes in methane ($CH_4$) concentrations and oxidation) at all latitudes." Since the manuscript already explains why tropopause region temperatures and ACE-FTS $N_2O$ are used simultaneously in the regression, no changes have been made other than the added statement detailed above.*

Reviewer 2

Compared with the previous version, it is significantly improved. I have still only a few minor comments listed below:

Introduction:

L36: At higher altitudes and more poleward latitudes, H2O variations tended to follow those of the modeled temperatures with a lag of a few months (tape-recorded effect, Mote et al., JGR, 1996).

*"(tape-recorded effect, Mote et al., 1996)" has been added.*

L48: A number of studies... Please include the Tao et al. 2023 paper, which mainly analyzes the interaction between temperature variations in the tropical tropopause region and changes in the Brewer-Dobson circulation. You use it later on in the paper. A detailed analysis of the changes in

methane concentrations is the biggest novel advantage of your paper. I would recommend reformulating this part.

*We now include the Tao et al. reference here as well.*

*At the end of this paragraph, we now also include the statement, "ACE-FTS is in a unique position when it comes to investigating the influence of $CH_4$ oxidation on stratospheric $H_2O$ trends as it is currently the only Earth observing satellite instrument that makes vertically resolved measurements of both $H_2O$ and $CH_4$ throughout the stratosphere."*

Fig. 12 and 13: Please switch.

*These figures are now in the correct order.*

Fig. 1 caption: ...where only regressing to linear components... I would not mention "linear components." You do not use it in the main text explaining your procedure like in lines 178-179. Linear components are trends. Please check other captions as well (e.g., Fig. 4).

*"...regressing to linear components and..." has been removed here and in the text.*

You have 15 figures in total. I would recommend shifting a few of them to an appendix.

*We have made an appendix and have moved two figures (time-lagged ENSO and trop figure and effect of correlation with AO figure), as well as their discussion, to there.*

Fig. 5: The optimal time lags scatter very strongly and do not look very convincing. I would shift it to the appendix. Their contribution is very weak.

*These were added in response to one of the previous reviewers. The figure and its discussion have been moved to the appendix.*

Fig. 8: Please include the abbreviation T_lag into the caption and maybe into the legend of the the color bar. By the way, its distribution is much more convincing than time lags of ENSO and Trop.

*We have added $T_{lag}$ to the caption.*

Fig. 14: This figure could also be shifted to the appendix. I did not get the point of the analysis with and without AO.

*This was added in response to one of the previous reviewers. The figure and its discussion have been moved to the appendix.*